# Wettability Studies of Capillary PTFE Membranes Applied for Membrane Distillation

**DOI:** 10.3390/membranes13010080

**Published:** 2023-01-08

**Authors:** Piotr Woźniak, Marek Gryta

**Affiliations:** Faculty of Chemical Technology and Engineering, West Pomeranian University of Technology in Szczecin, ul. Pułaskiego 10, 70-322 Szczecin, Poland

**Keywords:** brine, capillary PTFE membrane, membrane distillation, membrane wettability, oily wastewater

## Abstract

In the present study, the membrane distillation (MD) process was studied with the use of commercial polytetrafluoroethylene (PTFE) capillary membranes. For this purpose, both solutions with NaCl concentrations up to 300 g/L and brines contaminated with oil (70–120 mg/L) were used as feeds. The membrane’s wetting resistance was tested by conducting long-term experiments for over 3500 h. Using detailed studies, it has been shown that increasing the salt concentration from 35 to 300 g/L resulted in a 50% reduction in the permeate flux. Nevertheless, the membranes retained 100% of the salt rejection after 2000 h of the module’s operation. Moreover, it has been found that performing the MD process with brines contaminated with oil (120 mg/L) led to the wetting of some membranes pores, which it turn resulted in an increase in the distillate’s conductivity to 450 µS/cm after 700 h running the process. The mechanism of pore wetting by oil droplets adsorbed on the membrane’s surface was presented. Finally, the proposed method of membrane cleaning with the use of a solvent allowed restoring the initial MD module’s performance. Consequently, both the permeate flux and distillate conductivity were stable during the MD of the feed containing 35 g/L of NaCl over the next 280 h.

## 1. Introduction

For many years, there has been considerable interest in the use of hydrophobic porous polytetrafluoroethylene (PTFE) flat-sheet membranes in the membrane distillation (MD) process [1]. Initially, the above-mentioned membranes were manufactured for microfiltration (MF) purposes. Currently, they are constantly being improved, and as a result, they are successfully used in commercial MD spiral wound modules [2,3]. In such installations, a net-type spacer separates the two membranes and creates a channel for feed flow. Nevertheless, due to the high hydraulic resistance, low feed flow values are used, and this consequently limits the module’s performance to the level of 1–2 L/m^2^h [4]. It is worth noting that higher flows and, hence, greater process performance can be obtained by using capillary modules [5]. However, the supply of commercial capillary PTFE membranes, contrary to flat-sheet membranes, is limited [6,7].

Capillary PTFE membranes are mainly prepared using a cold-pressing method, including paste extrusion, stretching and sintering [6,7,8,9]. More specifically, this method is based on forcing a mixture of PTFE powder and paste lubricants through an extruder and forming a flexible capillary by using the paste extrusion process. In the subsequent stage, the PTFE capillary is annealed and stretched, which creates walls with a porous structure. It has been documented that both the annealing temperature and stretching have key impacts on the properties of the prepared capillary PTFE membranes [10]. For instance, the beneficial properties were obtained by increasing membrane lengths more than twice [11]. In turn, in [6], for the membrane fabricated at a stretching ratio and temperature of 2.4 and 50 °C, respectively, the reported NaCl rejection was equal to 99.99%. The stretching conditions determine both the pore’s size and pore size distribution, which has a significant impact on the liquid entry pressure (LEP) value obtained. For capillary PTFE membranes, LEP values in the range from 0.15 to 0.47 MPa were noted [6,11].

The wettability of the hydrophobic membrane’s pores is one of the fundamental issues impeding the implementation of the MD process. If the hydraulic pressure inside the MD module is lower than the LEP value, the distilled water does not wet the membranes [11,12,13], which was confirmed in three-year MD studies [12]. However, if hydrophilic groups form on the membrane’s surface as a result of polymer matrix degradation or the formation of deposits (fouling/scaling), the feed may flow into the pores that are in contact with them (surface wetting) [8,12]. This may initiate internal scaling, which causes wetting of the pores inside the wall; as a result, in some places, the wetted pores connect with each other, creating channels for feed leakage (partial wetting) [14]. This phenomenon often occurs as a result of salt crystallization on the membrane’s surface [14,15]. The crystals formed in the pores allow the feed to flow into the next pores. Importantly, this process is rarely reported in the literature. For this reason, several thousand hours of research was carried out in the work, which allowed indicating the suitability of the tested PTFE membranes for industrial applications.

It is well known that oil and gas extraction involves the formation of substantial amounts of produced water [8,13,16,17]. In addition to the rich mineral composition, such brines contain large amounts of petroleum pollutants, which are significant threats to the environment. For this reason, methods of treating the oily saline wastewaters are attracting remarkable interest. Among them, the MD offers several significant advantages [8,18,19,20,21,22]. A challenging area in the field of the MD process’s efficiency is preventing the phenomenon of membrane wetting, which occurs due to the adsorption of organic components on the hydrophobic membrane’s surface. The wetting of hydrophobic PTFE membranes during the separation of oily water has been reported in the literature [8,19,23]. Summarizing the literature data, it can be concluded that, generally, this phenomenon is prevented by applying a thin hydrophilic layer to the membrane’s surface, which ensures oleophobic properties [16,23,24,25]. For instance, Yu et al. [8] demonstrated that the suitable resistance to wetting by oily wastewaters can be obtained by coating PTFE capillaries with a thin layer of Nafion membrane. Therefore, the surface properties of the MD membrane applied for the separation of oily water are different from those developed for water desalination, where it is important to increase the hydrophobicity of the membranes. This can be achieved by using various methods, such as the manipulation of pore structures and hydrophobic decorations [26]. Furthermore, wetting the hydrophobic membranes by using hydrophobic oil should not facilitate the entry of water into the pores. The paper presents a model explaining this phenomenon.

It is well known that hydrophobic membranes may be wetted during the continuous separation of feed containing large amounts of oils (e.g., 1–2 g/L). In the literature, there is general agreement that the membrane’s wetting is a main weakness of the MD process, which significantly limits its application in oily wastewater treatments. The MD process is mainly proposed for brackish and seawater desalination [1,2,3,27]. However, it should be pointed out that seawater may be temporarily contaminated by oil due to the bilge water discharge by ships [28,29,30]. Usually, in such cases, the oil concentration is much lower than that noted in produced water, and contamination occurs in the form of an emulsion containing oil droplets with a diameter below 50 µm [13,31]. The previously performed investigation [32] demonstrated that polypropylene (PP) membranes show wetting resistance by such solutions. With regard to PTFE membranes, in [8], it was observed that the wetting resistance increased with decreasing oil concentrations in the feed from 1000 to 100 mg/L. It can be attributed to the fact that generally, lower concentrations are characteristic for stable emulsions. In turn, in the current study, the resistance to wetting by oil emulsion with respect to commercial PTFE membranes is presented.

Although significant achievements in preparing modified capillary PTFE membranes have been documented in recent years [6,7,8,10,11,33], up until now, for the implementation of the MD process, capillary modules with commercially available membranes are tested. There are only a few manufacturers producing membranes that can be used in the construction of commercial modules. It is essential to mention that in view of industrial purposes, investigations with respect to membrane properties during the long-term process are particularly important. However, to the best of the authors’ knowledge, this issue has not yet been fully evaluated. For this reason, the challenge is to indicate PTFE capillary membranes that are suitable for the construction of a pilot plant. Noteworthily, testing this type of membrane is essential for the further development of the MD process. Taking the abovementioned into account, herein, the suitability of the capillary PTFE membranes manufactured by an Indian company for long-term MD processes was assessed. In the present work, the possibility of the industrial application of these membranes was assessed by testing their resistance to wetting in over 3500 h of research with the use of brines contaminated with oil as a feed.

## 2. Materials and Methods

Generally, in the field of wettability studies, the fundamental characteristic of the membrane is the structure (porosity and pore diameter), which for a given value contact angle prevents the aqueous solution from entering dry pores until the operating pressure (P_F_) exceeds the water liquid entry pressure (LEP_w_) [6,10,11]. However, it should be mentioned that during the long-term membrane operation, pore wetting may occur if P_F_ << LEP_w_, which is caused by slow changes in the properties of the membrane’s surface or the fouling phenomenon [6,8,10]. In the present study, in order to investigate the effect of these factors on the wetting of the capillary PTFE membranes, the impact of P_F_ was eliminated by using a submerged module (Figure 1). The method of attaching the capillary membranes to the module is shown in Figure 1b.

In the current study, long-term MD experiments with the use of the commercial capillary PTFE membranes manufactured by TECH INC (Tamil Nadu, India) are presented. The membranes were supplied as a bundle of 50 capillaries that are 30 cm long. The capillaries used for the MD test were pulled randomly from the bundle. The capillary modules were made by gluing the ends of two capillaries to glass tubes (Figure 1b). For the MD study, three similar modules (M#1–3) with an active length of 22 cm (external area 33 cm^2^) were prepared.

During the MD tests, distilled water at the temperature of 22 ± 1 °C was flowing (150 mL/min) inside the capillaries. The membranes were immersed in a feed heated with an RCT Basic magnetic stirrer (IKA-Werke GmbH, Staufen, Germany) to the temperature of 50–80 ± 0.5 °C. Solutions with a NaCl concentration of 35–300 g/L were used as a feed. In order to investigate the separation of oily wastewater, a feed containing NaCl (300 g/L) and oil (70–120 mg/L) was prepared. Oil was collected from the bilge water surface in the ship’s engine room.

As a result of pore wetting, there is the possibility of wetting part of the wall through which salt diffuses from the feed into the distillate. In order to detect it, changes in the distilled conductivity were investigated. In addition, partial wetting and oil adsorption reduce feed evaporation surfaces, which reduces the permeate flux. To assess the intensity of these phenomena, the MD tests were run continuously. Once a day, the distilled conductivity and permeate flux were determined. The permeate flux (L/m^2^ h) was calculated as the mean of the distillate volume collected over 24 h, according to the following equation:J = dV/A t (1)
where dV (L) is an increase in the volume of distillate at time t (24 h), and A (m^2^) is the module area. The volume losses in the feed tank were replenished every day with distilled water.

Pore size and pore size distribution were determined using the Capillary Flow Porometer Porolux 500 (Porolux, Belgium) by using the gas–liquid displacement method. The gas was provided by nitrogen. Bubble point, smallest pore size, biggest pore size and pore size distribution were calculated and reported by the computer connected with the equipment. For the measurements, the membranes were wetted using the “Porefil 16.39” fluid with a surface tension of 16.4 mN/m.

In order to determine the oil concentration, the samples were extracted with solvent S316 (Horiba) and analyzed by the IR method using an OCMA 500 apparatus (Horiba, Kyoto, Japan). The distillate conductivity and TDS were measured with the use of the 6P Ultrameter (Myron L Company, Carlsbad, CA, USA). The membrane’s morphology was determined by an SEM microscope (SU8000), while the composition of the deposits formed on the membrane’s surface was analyzed using the SEM-EDS (Hitachi High Technologies Co., Tokyo, Japan). The devices and methodology have been described in detail in previous studies [32,34].

## 3. Results and Discussion

### 3.1. Membrane Performance

The structure of the capillary PTFE membrane used in the present study is shown in Figure 2. The inner diameter and wall thickness of virgin capillaries were equal to about 1.4 mm and above 0.4 mm, respectively (Figure 2a). The porous structure of the wall was created by small agglomerates (Figure 2b). On the wall’s surface (Figure 2c,d), the solid agglomerates of PTFE particles (node) interconnected by cracks called “fibrils” were observed, which is a characteristic structure for membranes obtained by the stretching method [6,11]. A slightly higher density of fibrils was formed on the inner surface (Figure 2d), which was also observed during the studies of other capillary membranes [6,10]. A significant difference in relation to the previously presented PTFE membranes in the literature was found in the structure of the inner wall. It consisted of numerous cylindrical PTFE agglomerates smaller than 0.5 µm in size connected by thin rods that are 0.1 µm in diameter (Figure 2b). The membranes presented so far had a node–fibril structure inside the walls that is similar in appearance to those presented on the inner surface; however, it had a greater proportion of fibrils [6].

It has been found that the wall thickness and the diameter of the capillaries showed slight fluctuations. Indeed, the average values of the abovementioned parameters determined for the three samples are shown in Table 1. The results obtained in the present study are in line with those presented in [10], wherein the fluctuations in the value of the capillaries’ wall thickness and significant distortions in the shape of the circular cross section have been demonstrated. Therefore, it can be clearly indicated that obtaining a homogeneous cross section of the capillary PTFE membrane is a great challenge. Indeed, some fluctuations in the wall dimension can be expected in membranes prepared by using the a cold-pressing method.

The determined pore size distribution is presented in Figure 3. It can be observed that the diameter of most pores was within the range from 0.2 to 0.4 µm; however, the size greater than 0.7 µm was also noted. According to results obtained by the bubble point method, the greatest pores size was equal to 0.745 µm (Table 2). Noteworthily, similar shapes with respect to pore size distributions were obtained for PTFE capillaries by Wang et al. [10]. Pores with the largest diameters are wetted the fastest [13]; hence, their presence may cause the partial wetting of the tested membranes [14].

SEM studies of the membrane samples from the M#1 module after the MD process run are shown in Figure 4. It can be observed that during the MD, the membranes changed their structure, which is also confirmed by the measurement results presented in Table 1. The most significant changes were observed in the structure of the inner wall (Figure 4b). Indeed, the distances between the nodes decreased significantly compared to those observed for the virgin membrane (Figure 2b). The dimensions of the fibrils on the wall’s surface also changed (Figure 4c,d). It can be inferred from this study that the long-term heating (60 °C) of membranes caused the shrinkage of their structure. As a result, the individual sizes of capillaries decreased by more than 14%. Moreover, the SEM cross-section analysis showed that although the capillary’s cross section was similar to that shown in Figure 2a for most of the samples, in some cases, the wall thickness was not uniform and varied within the range from 297 to 472 µm (Figure 4a).

Enlarged SEM images of fibrils structure are presented in Figure 5. It has been noted that as a result of long-term annealing processes, the thickness of the fibrils on the outer surface of the PTFE capillary (Figure 5b) was significantly thinner compared to those observed for the virgin membrane (Figure 5a). In turn, with regard to the inner surface (Figure 5c,d), there were only slight differences. It can be explained by the fact that this surface was in contact with the cold distillate (22 °C), which prevented thermal transformations. Noteworthily, significant changes in the shape and structure of fibrils as a result of annealing were also presented in a study [10]. It is probable that the tested membranes were annealed too briefly during their production to achieve full thermal stability.

MD is a phase change process; hence, the value of the permeate flux depends on the amount of energy supplied for the evaporation surface [35]. This, in turn, is determined by the convective heat transfer coefficient, the value of which results from the intensity of the feed mixing at the boundary layer [2,35]. Thus, in the present study, the impact of the rotational speed (50–700 rpm) of the stirrer heating the feed tank (Figure 1a) on the permeate flux was investigated (Figure 6). It has been noted that the process performance stabilized at a rotational speed above 500 rpm. Therefore, all experimental investigations were carried out at a rotational speed equal to 700 rpm.

The impact of the feed temperature on the process performance is shown in Figure 7. Obviously, increasing the temperature from 50 to 80 °C led to exponentially increasing the vapor pressure [7,35]. As a result, an increase in the permeate flux from 1.5 to 10 L/m^2^h was reported. Performing a comprehensive literature review shows that, at the feed temperature of 65 °C, flux at the level of 5 L/m^2^h was obtained for other commercial PTFE membranes with similar parameters [6,7]. In turn, a twice lower permeate flux was noted for capillaries covered with a thin layer of Nafion membrane, which was used to eliminate the wetting of PTFE by oily wastewaters [8].

In turn, the investigation on the impact of T_F_ on the M#1 and M#3 modules performance showed some differences in the permeate flux obtained (Figure 7). The reported results may be attributed to the observed differences in the structure of the capillary wall’s surfaces (Figure 5). The presented results were obtained for the initial period (40 h) of the module’s operation, and the differences in the flux value for a given temperature, T_F_, resulted from differences in the stabilization of the membrane’s structure. The change in the module’s performance as a function of operating time (Figure 8) showed that the membranes stabilized for more than 150 h of the MD process, and subsequently, 98% of the initial permeate flux was noted. In each of the process’s studies, distillate conductivity did not exceed 2 μS/cm, which indicated that the membranes retained their non-wettability during this period. It was shown that not only the pore size but also the diameter of the inner wall of the fibers has a significant impact on the transport properties of hydrophobic membranes [26]. Therefore, the virgin membrane’s change (Figure 2b) to a more compact structure (Figure 4b) during the MD had an impact on the permeate flux.

Small differences in the permeate flux could also be due to the wall’s thickness fluctuations (Figure 4a). Its thickness affects not only mass transport resistance but also the amount of Q_C_ heat conducted from the feed to the distillate, which is described by Equation (2):(2)$$Q_{C}=\frac{{\varepsilon \;\lambda }_{V}+\left(1-\varepsilon \right){\;\lambda }_{\mathrm{PTFE}}}{S_{m}}\;\left(T_{1}-T_{2}\right)$$
where ε denotes membrane porosity, S_m_ denotes membrane thickness, T_1_ denotes the temperature of the evaporation surface, T_2_ denotes the temperature of the condensation surface and λ_V_ and λ_PTFE_ denote the conductive heat transfer coefficient of vapor (gas phase) and the material of the membrane matrix (PTFE), respectively. The interfacial temperatures, T_1_ and T_2_, are different from the measured bulk temperatures on the distillate and feed side, and this phenomenon is called temperature polarization [2,36].

An advantage of submerged MD modules is maintaining a constant feed temperature outside the membranes. However, the temperature of the distillate stream flowing inside the capillaries decreases along the module’s length, which decreases the permeate flux. This is mainly due to the increase in the temperature of the condensation surface (T_2_). Using the MD model presented in [36], the changes in the T_2_ value for different wall thicknesses were calculated (Figure 9). In each of the analyzed cases, the reduction in wall thickness resulted in a faster increase in the temperature of the boundary layer on the distillate side. However, the permeate flux changes calculated for different feed temperature values did not exceed 5%, which is much less than the differences presented in Figure 7. This shows that the main reasons for the differences in the permeate flux were the changes in the wall’s structure (Figure 4) and, to a lesser extent, the fluctuations in its thickness.

It should be pointed out that the changes in the membrane’s properties observed during the investigations of the MD process concerned not only the membrane’s morphology but also the contact angle (Figure 10). The contact angle determined for the virgin membranes was equal to 134.4 ± 1.2°. It is important to mention here that this result is in line with that reported in a previous study [10], wherein a slightly lower value of the contact angle (130°) was observed for the capillary PTFE membrane. It has been found that after 2000 h of the MD process run, the contact angle was equal to 81 ± 1.5°, which clearly indicated that the surface of the PTFE membrane used in the present study was wetted. However, the pores inside the membrane’s wall were not wetted, and as it is presented in the next section, even for feed containing 300 g NaCl/L, almost 100% of salt retention was obtained, which is characteristic whenever surface wetting occurs [14].

### 3.2. Separation of Concentrated NaCl Solutions

After a period of stabilizing the membrane’s properties (Figure 8), the membranes provided stable permeate flux (Figure 11). Indeed, flux measuring 3 L/m^2^h (M#1 module) during 1000 h of the NaCl solution’s (35 g/L) separation was noted. A slightly greater performance was obtained for the M#3 module. Therefore, it can be expected that industrially produced membranes may vary in terms of the permeability achieved. However, several hundred capillaries are mounted in the MD module, which gives an average value, and the performance of the industrial modules is similar.

Despite long-term module exploitation, the distillate’s conductivity was below 5 µS/cm, which indicated that the membranes retained their non-wettability during the separation of the NaCl solution (35 g/L).

After 1000 h of the M#1 module’s operation, the NaCl concentration in the feed gradually increased from 35 to 300 g/L (Figure 12). For the salt concentration of 200 g/L, the permeate flux decreased to 2.5 L/m^2^h, which indicated a 16% decrease in the permeate flux, similarly to that presented by Su et al. [7]. In turn, a much greater decline in the process’s performance (30%) for twice lower concentrations (100 g/L) was demonstrated in [11], probably due to the less favourable hydrodynamic conditions leading to more significant polarization effects. It can be clearly observed that when the salt concentration in the feed increased to 300 g/L, the permeate flux significantly decreased to 1.8 L/m^2^h. Importantly, despite such a high salt concentration, distillate conductivity increased only to 17 µS/cm, which proved almost 100% of salt retention. It must be stressed that the final module performance was stable for over 200 h. The experimental results reported above show that even for almost saturated brines used as a feed, the tested PTFE membranes were not wetted. The decrease in the contact angle value from 134 to 81° shown in Figure 10 indicates that during the long-term period, the surface wetting of the studied membranes occurred. The wall thickness of the membranes was about 400 μm; therefore, despite the wetting of the PTFE capillaries’ surface pores, a layer of pores filled only with the gas phase was still preserved inside their walls. Such a property makes it possible to meet the basic condition for the correct operation of the MD process.

A concentration of 300 g NaCl/L approached saturation, which may cause scaling. In this case, the separation possibility of the saturated solutions can be achieved by reducing the salt concentration in the feed by using a crystallizer attached to the MD installation (membrane distillation crystallizer) [15,37].

### 3.3. Separation of NaCl Solutions Contaminated by Oil

The results presented and discussed in the previous section (Figure 12) showed that the PTFE capillary membranes were not wetted by the solution with the NaCl concentration of 300 g/L. This noteworthy observation indicates the possibility of obtaining very high water recovery rates in the MD process and the possibility of implementing zero liquid discharge (ZLD) technology [38,39,40,41].

An important issue that must be considered is the oil adsorption on the PTFE membrane’s surface. For this reason, in a subsequent step of the studies, the PTFE membrane’s resistance to oily contamination was thoroughly investigated. For this purpose, an oil emulsion was added to the feed containing 300 g/L of NaCl, maintaining oil concentrations within the range of 70–120 mg/L during the experimental investigation. The obtained results are shown in Figure 13. The addition of oil (120 mg/L) to the feed resulted in a systematic decrease in the module’s performance after 50 h. Finally, after 700 h, the permeate flux decreased from 1.9 to 0.8 L/m^2^h and remained stable until the end of the process run. With regard to distillate conductivity, a significant increase was noted after 500 h; consequently, it was equal to 450 µS/cm (TDS 0.2 g/L) at the end of the process run. Undoubtedly, this finding indicated partial wetting, which resulted in feed leakage into the distillate. However, for the feed concentration of 300 g/L, high salt retention (>99%) has been recorded. It is probable that only a few of the largest pores (Figure 3) were wetted.

The obtained results confirm that the presence of oil in the feed contributes to the pore’s wetting process. However, this is strongly affected by the level of oil concentration. Indeed, Sha et al. [8] have shown that the addition of oil (500 ppm) to a 3.5% NaCl solution resulted in complete membrane wetting after less than 10 h of the MD process run. For higher concentrations (e.g., 1000 mg/L), a rapid decrease in the process’s performance was observed due to blocking of membranes caused by oil adsorption [25]. Different results were obtained for low oil concentrations, and the low degree of membrane wetting was observed during the 700 h of the separation of the oily feed containing 70–120 mg/L (Figure 13). Similarly, the high resistance relative to oil contamination was demonstrated for PTFE membranes used for the treatment of wastewater with low oil concentrations from a petrochemical industry using a submerged MD bioreactor [42]. These results indicate that the temporary emergency supply of MD installations by seawater contaminated by, e.g., bilge water (low oil concentration) should not be an operational difficulty.

Presented and discussed in the literature, different intensities of the wetting phenomenon of hydrophobic membranes by oils most probably resulted from differences in oil dispersions in the feed. As it was shown in [21,32], the size of oil droplets and the stability of the emulsion have decisive influences on the membrane’s wetting phenomenon. This phenomenon is schematically shown in Figure 14a. In the emulsion, large droplets are less stable; thus, they can be deposited on the membranes and wet the pores (Figure 14a—pore D), which blocks the vapor’s flow [25]. Moreover, the decrease in the contact angle value (Figure 10) indicates that the water adsorption on the PTFE capillaries’ surface occurs during the MD process. A similar phenomenon was found for polypropylene (PP) membranes, which significantly reduced oil fouling [21,32]. As a result, the oil droplets do not block the pores, as shown in Figure 14a (pores A).

The adsorption of water on the hydrophobic membrane’s surface significantly depends on the module’s lifetime [43]. Changes in the properties of PTFE membranes are presented in Figure 8, which indicate that the membranes stabilized for over 150 h during the MD process run. The PP membranes showed a similarly long stabilization period, after which a slight increase in the hydrophilicity of the membrane’s surface was found, which allowed reducing the oil fouling phenomenon [21]. Therefore, it is advantageous if new MD modules are supplied with clean water (without the oil) during the initial 2–3 days, which obtaining a membrane surface with hydrophobic–hydrophilic properties [32]. Omitting the stabilization period meant that the oil droplets completely blocked the pores of PTFE membranes after only 20 h of the MD process [25].

The tested PTFE membranes had large pores on the surface (Figure 5) into which the feed can flow (Figure 14a—pore B). As water evaporates, the oil droplets (pore C) thicken and the salt concentration increases, which reduces the thickness of an electric double layer. The decrease in the double-layer thickness reduces the influence of the electrostatic repulsion, promoting the coalescence [44]. As a result, oil fills the pore and closes the feed inside it (Figure 14a—pore E). A similar result can also be obtained by the adsorption of a large oil drop on the membrane’s surface.

Closing even a small amount of brine by oil in the pore can initiate internal scaling and, as a result, cause membrane partial wetting, which is schematically shown in Figure 14b. Due to water evaporation, the brines become supersaturated, which causes salt crystallization. The fact that the feed has lower concentrations results in osmotic transports through the oil from the feed to the brine in the pores. In [45], using a PTFE capillary partially filled with oil, it was shown that oil acts as a semipermeable membrane for water transport when separating the brines of different saline. Feed transport as a water-in-oil emulsion is also possible. The crystallization of salt in the pore allows the brine to move inside the pores until it fills them completely. Wetting the pores releases the brines into the distillate, resulting in an increase in distillate conductivity. Such a wetting process proceeds very slowly; therefore, during the tests, a greater increase in distillate conductivity occurred only after 400 h MD (Figure 13).

It is well known that the repeated MD process may result in significant fouling phenomenon, e.g., caused by membrane contamination with oil [21]. Consequently, the recovery of the membrane’s performance is required [42]. The efficiency of membrane cleaning processes after the MD process run (Figure 13) is shown in Figure 14.

It has been found that replacing the brine with water for 48 h led to slight increase in the M#3 module’s performance. Thus, it can be clearly stated that water (60 °C) does not ensure the effective removal of petroleum contamination from the surface of PTFE membranes. In the subsequent step, the membranes were naturally dried (Figure 15, point D), which allowed increasing the permeate flux from 1 to 1.9 L/m^2^h (feed: 35 g/L NaCl). This finding shows that the observed decrease in the process’s performance resulted not only from oil adsorptions on the membrane’s surface but also from the wetting of some membranes pores. Nonetheless, during the next 72 h of the MD process run, flux was reduced to 1.4 L/m^2^h. During the drying of the wetted pores, salt crystals are formed; consequently, they dissolve when the MD process resumes. This phenomenon facilitates the wetting of the pores and leads to a decrease in the permeate flux. This confirms that drying the wetted membranes is not an effective method for their regeneration. Finally, a stable increase in the permeate flux to the level of 2.4 L/m^2^h was obtained after cleaning the membranes with the use of solvent S316, which was applied by the Horiba company for very effective oil dissolving processes. Its hydrophobic properties also cause a thorough displacement of brines from the pores. Hence, it can be concluded that removing oil and brines from the membrane’s wall prevents the pores from re-wetting. Organic solvents can cause membrane swelling; therefore, it is necessary to choose solvents suitable for PTFE.

## 4. Conclusions

In the present study, the results of MD process studies conducted for over 3500 h were presented. It has been demonstrated that the capillary PTFE membranes are suitable for the water desalination and brine concentration. Indeed, during the entire investigation period, the membranes retained their non-wettability and the salt retention was close to 100%, even for the feed containing 300 g/L of NaCl.

Moreover, it has been determined that intensive mixing in the feed tank (700 rpm) ensured the elimination of salt crystallization on the membrane’s surface, despite the MD process (60 °C) of the solution occurring with salt concentrations close to saturation (300 g/L).

By conducting detailed studies, it has been shown that the addition of a small amount of oil (70–120 mg/L) to the NaCl solution (300 g/L) caused oil adsorption on the membrane’s surface, which consequently led to a decrease in the permeate flux by more than 50%, with a simultaneous increase in the distillate conductivity to 450 µS/cm. This result confirms that the long-term contact of the PTFE membrane with the oily contaminants resulted in pore wetting. However, cleaning the membranes with the use of solvent S316 has proven to be an effective method for restoring the process’s performance.

Oil acts as a semipermeable membrane for water transport when separating the brines of different saline; hence, its adsorption on the membrane’s surface does not limit the feed transport inside the pores, which may cause internal scaling and result in partial membrane wetting.

Finally, the results obtained in the present study demonstrated that the long-term contact of membranes with hot feeds led to changing the membrane’s structure and their shrinkage by 15%. However, the observed phenomenon led to only a 2–3% decrease in the initial process performance. Hence, it can be concluded that the membranes used in the present study can withstand the conditions of the MD process and can be used for the construction of capillary MD modules.

## Figures and Tables

**Figure 1 membranes-13-00080-f001:**
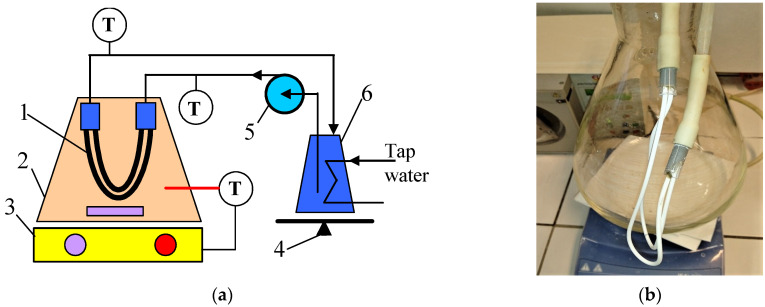
Experimental MD setup. (**a**) Installation scheme: 1—submerged module; 2—feed tank; 3—magnetic stirrer with heating plate; 4—balance; 5—peristaltic pump; 6—distillate tank; T—thermometer. (**b**) Submerged module connected with the distillate line.

**Figure 2 membranes-13-00080-f002:**
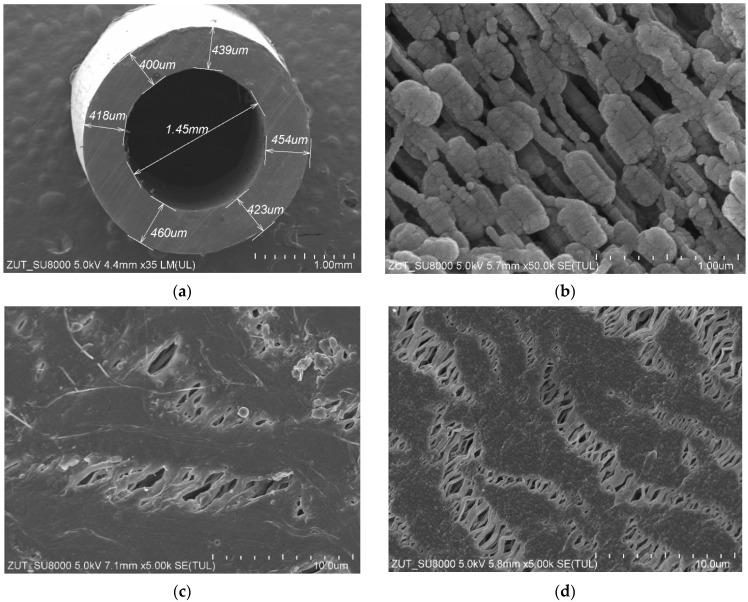
SEM images of the virgin capillary PTFE membrane: (**a**) capillary cross section; (**b**) wall cross section; (**c**) external surface; (**d**) internal surface.

**Figure 3 membranes-13-00080-f003:**
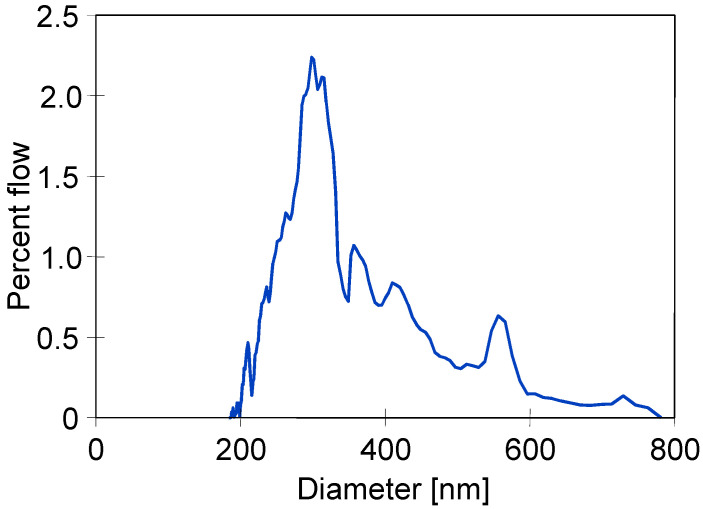
Pore size distribution of the capillary PTFE membrane used in the present study.

**Figure 4 membranes-13-00080-f004:**
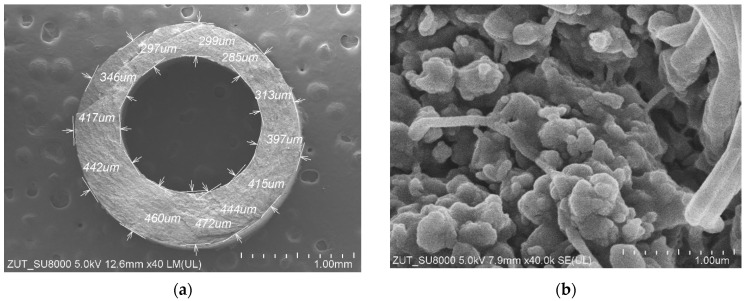
SEM images of the capillary PTFE membrane collected from the M#1 module after 2000 h of the MD process run (T_F_ = 60 °C): (**a**) capillary cross section; (**b**) wall cross section; (**c**) external surface; (**d**) internal surface.

**Figure 5 membranes-13-00080-f005:**
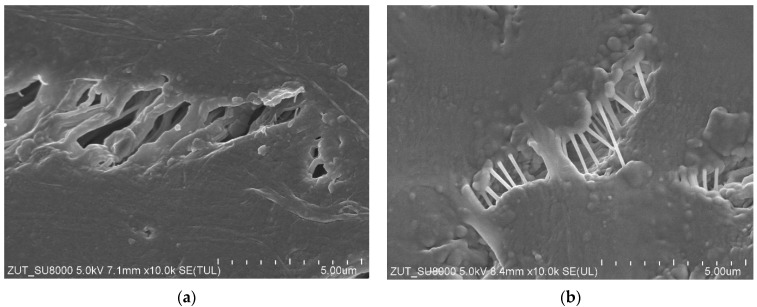
SEM images of the node–fibril structure for virgin and PTFE membrane used for the MD process (2000 h). External surface: (**a**) virgin; (**b**) after MD. Internal surface: (**c**) virgin; (**d**) after MD.

**Figure 6 membranes-13-00080-f006:**
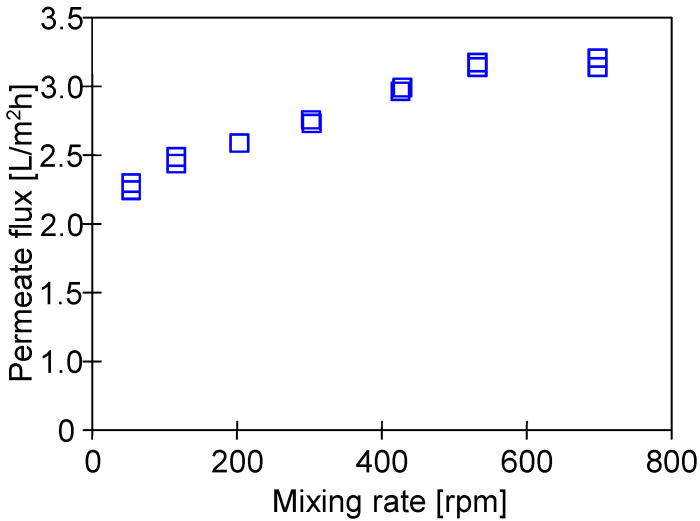
The impact of the stirrer rotational speed on the M#3 module. Feed: NaCl solution (35 g/L). T_F_ = 60 °C.

**Figure 7 membranes-13-00080-f007:**
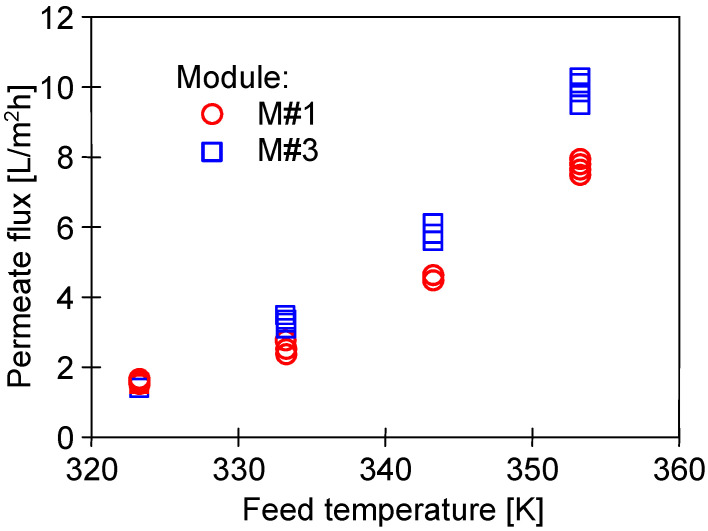
The impact of feed temperature on the permeate flux. Feed: NaCl solution (35 g/L).

**Figure 8 membranes-13-00080-f008:**
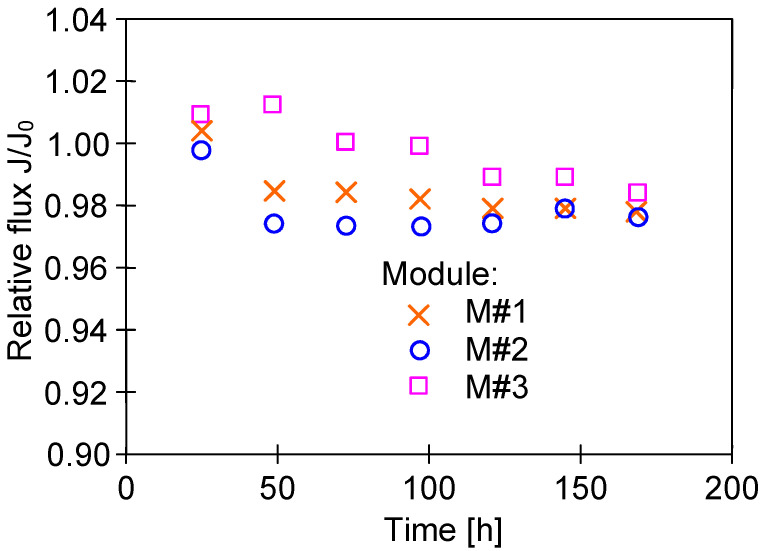
Changes in the relative flux during the initial period of the MD process run. Feed: NaCl solution (35 g/L), T_F_ = 60 °C.

**Figure 9 membranes-13-00080-f009:**
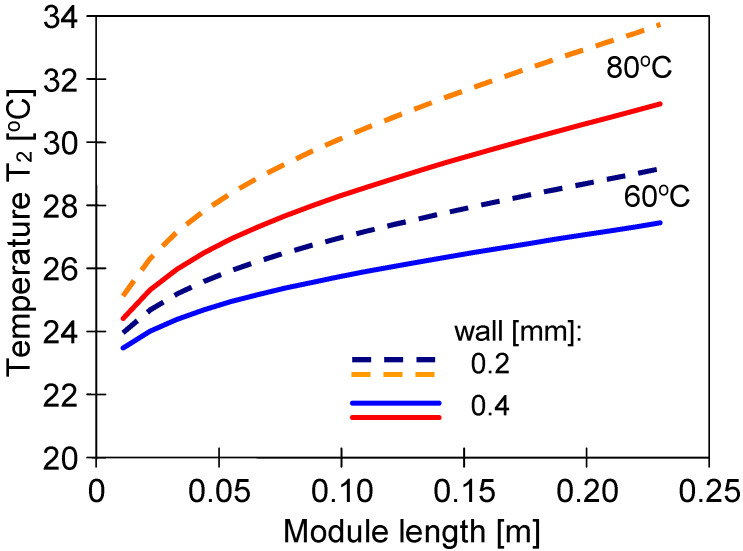
Changes in the interfacial surface temperature (T_2_) on the distillate side calculated along the module’s length for feed temperatures 60 °C and 80 °C. Distillate inlet temperature T_D_ = 22 °C. Membrane wall thickens at 0.2 and 0.4 mm.

**Figure 10 membranes-13-00080-f010:**
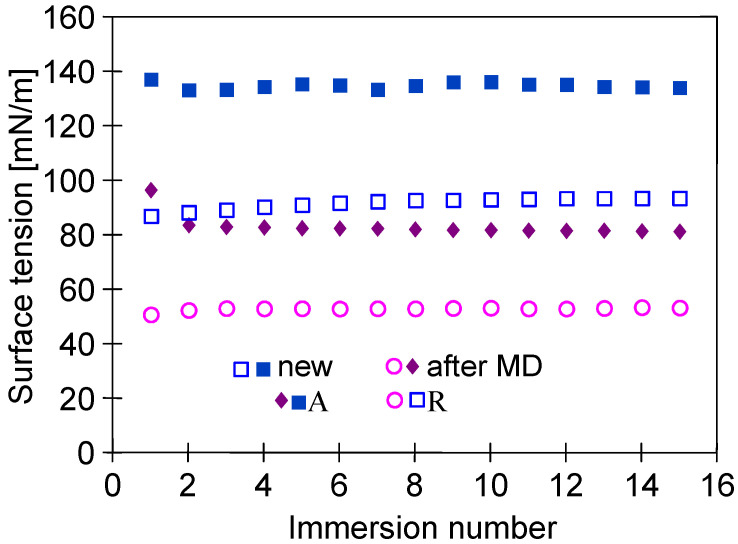
Changes in advancing (A) and receding (R) angles measured by the Wilhelmy plate method. Samples: new—virgin membrane; after MD—membrane collected from module M#1 after 2000 h of MD process.

**Figure 11 membranes-13-00080-f011:**
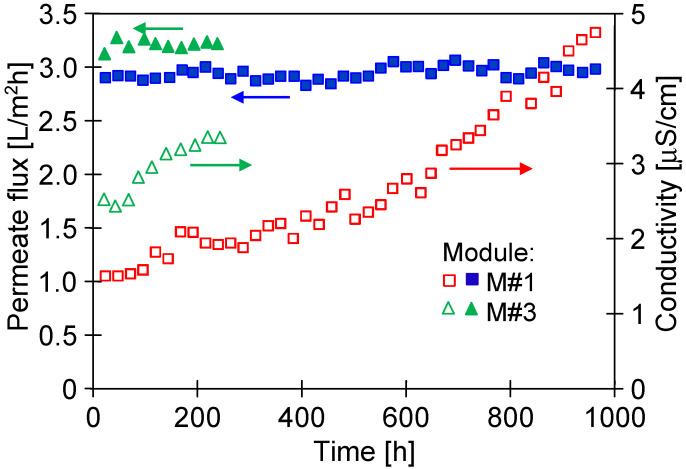
Changes in the permeate flux and distillate conductivity during the MD process. Feed: NaCl solution (35 g/L). T_F_ = 60 °C, modules M#1 and M#3.

**Figure 12 membranes-13-00080-f012:**
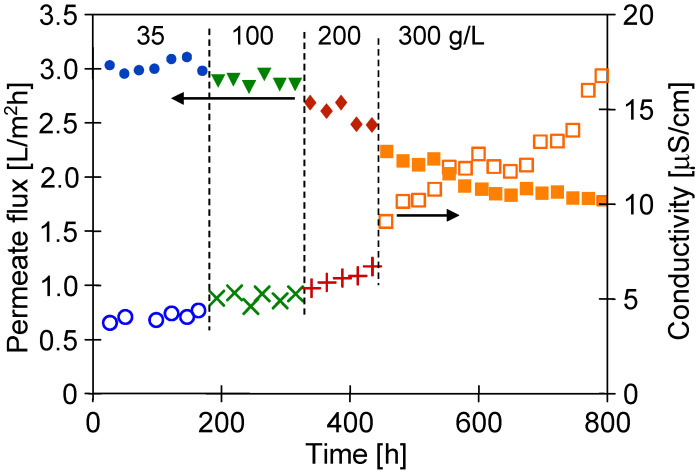
Changes in the permeate flux and distillate conductivity during the MD process of NaCl solutions (35–300 g/L). T_F_ = 60 °C, module M#1.

**Figure 13 membranes-13-00080-f013:**
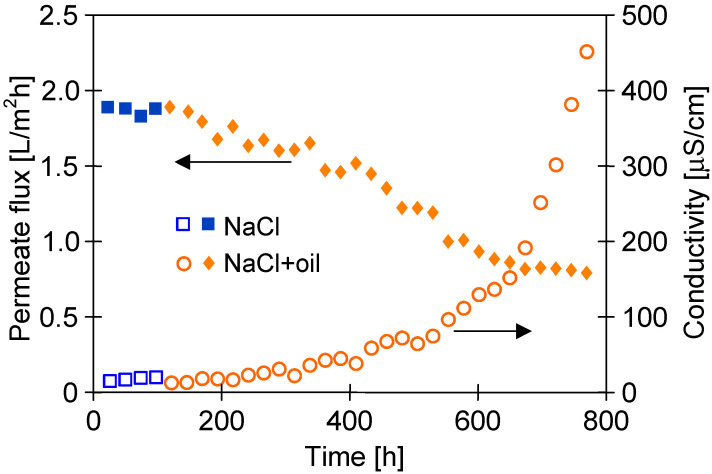
Changes in the permeate flux and distillate conductivity during the MD process of the NaCl solution (300 g/L) contaminated by oil (70–120 mg/L). T_F_ = 60 °C, modules M#3.

**Figure 14 membranes-13-00080-f014:**
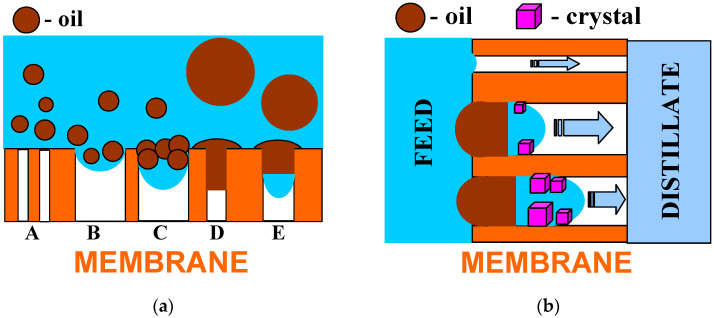
Model of hydrophobic membrane wetting during the MD process of brine contaminated with oil. (**a**) Influence of pore size and oil droplet’s diameter; (**b**) effects of internal scaling.

**Figure 15 membranes-13-00080-f015:**
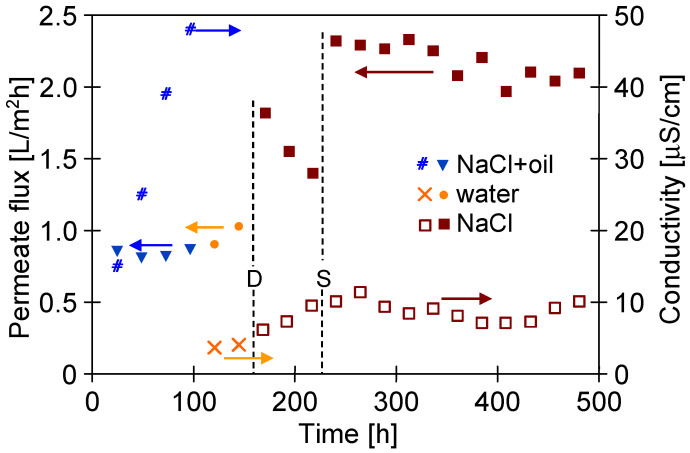
Changes in the permeate flux and distillate conductivity during the MD process of NaCl solution (300 g/L) contaminated by oil (70–120 mg/L). T_F_ = 60 °C, module M#3. Points: D—membrane drying; S—membranes rinsed by S316.

**Table 1 membranes-13-00080-t001:** Dimensions of capillary PTFE membranes used in the present study.

Membrane	Internal Diameter (μm)	Wall Thickness (μm)
virgin	1450 ± 5	440 ± 20
after MD process	1240 ± 21	378 ± 69

**Table 2 membranes-13-00080-t002:** Dimensions of capillary PTFE membranes used in the present study.

Bubble Point Pressure (bar)	Bubble Point Pore Size (μm)	Smallest Pore Pressure (bar)	Smallest Pore Size (μm)
0.88	0.745	3.260	0.201

## Data Availability

Not applicable.

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
