# Peer review of "Wettability Studies of Capillary PTFE Membranes Applied for Membrane Distillation"

_membranes, 2023, doi:10.3390/membranes13010080_

Round 1
Reviewer 1 Report
The paper entitled ''Wettability Studies of Capillary PTFE Membranes Applied for Membrane Distillation'' aims to investigate the wetting effect in a submerged membrane for oily wastewater treatment in MD. The study is interesting and the manuscript is well-written. The authors used commercial hydrophobic membranes in MD experiments. Feed water contained NaCl (30-300 g/L) and oil (70-120 mg/L|) was used. In my opinion, the paper can be accepted for publication after a revision. My comments are as follows.
1. The experiment duration should be given as days, not hours (please fix this in all the text and figures).
2. Abstract, Line 18: The statement ''the MD process efficiency'' is unclear to me. Please rephrase it with a clear statement.
3. In the introduction section, please mention the wetting mechanisms in MD in detail and highlight the novelty of the study.
4. In the last paragraph of the introduction, please mention the aim of the study.
5. Please give the temperature as ºC, not K.
6. Please present the specifications of M1, M2 and M3 membranes in Materials and Methods.
7. What was the Delta T in the experiments?
8. What was the effect of the capillary thickness (wall dimension) on heat and mass transfers?
9. Permeate flux values look too low. What was the reason for the low permeate flux?
10. Please explain whether wetting can be reduced with membrane cleaning (Lines 318-319).
11. Wetting includes different forms, surface, partial and full wetting. Please define the wetting observed in the study according to these wetting types.
Author Response
Dear Reviewer,
I would like to express my sincere gratitude to the Reviewer for the interest in my work and the valuable comments and constructive suggestions.
As indicated below, I have taken into considerations all comments provided by the Reviewer and I have made changes and corrections accordingly to the indications.
Thank you for your time and effort.
Yours sincerely,
Marek Gryta
The paper entitled ''Wettability Studies of Capillary PTFE Membranes Applied for Membrane Distillation'' aims to investigate the wetting effect in a submerged membrane for oily wastewater treatment in MD. The study is interesting and the manuscript is well-written. The authors used commercial hydrophobic membranes in MD experiments. Feed water contained NaCl (30-300 g/L) and oil (70-120 mg/L|) was used. In my opinion, the paper can be accepted for publication after a revision. My comments are as follows.
- The experiment duration should be given as days, not hours (please fix this in all the text and figures).
- I suggest leaving hours - as a result of converting to days in the article, we would have statements like: "after 2 days of MD, the wetting was similar to that after 0.42 d (i.e. 10 h) observed in another work". Most MD papers use hours, so comparing the results would force the reader to convert days to hours, which makes the analysis of the results difficult. We have a similar situation with permeate flux, most membrane papers use [L/m2h] but some authors use [m3/m2s]. As a result, you need to use a calculator to understand and compare their results.
In addition, if we have good, resistant membranes (as in this case), the changes in the following days are small and the average result from 24 h of measurements is similar to what was e.g. after 5 or 20 h. Therefore, it is possible to analyze not only the results with a step of 24 h, e.g. 24 - 48 - 72 etc, but also as 20-50-100 etc.
To emphasize this in Experimental I added:
L146
Once a day, the distilled conductivity and the permeate flux were determined. The permeate flux (L/m2h) was calculated as the mean of the distillate volume collected over 24 h, according to the following equation:
J=dV/A t (1)
where dV (L) is an increase in the volume of distillate at time t (24h), and A (m2) is the module area. The volume losses in the feed tank were every day replenished with distilled water.
- Abstract, Line 18: The statement ''the MD process efficiency'' is unclear to me. Please rephrase it with a clear statement.
It was corrected:
L.18 “with the use of a solvent allowed to restore the initial MD module performance.”
- In the introduction section, please mention the wetting mechanisms in MD in detail and highlight the novelty of the study.
These aspects have been clarified, new descriptions have been added:
L.50
The wettability of the hydrophobic membranes pores is one of the fundamental issues impeding the implementation of MD process. If the hydraulic pressure inside the MD module is lower than the LEP value, distilled water does not wet the mem-branes, which was confirmed in three-year MD studies [12]. However, if hydrophilic groups are formed on the membranes surface as a result of polymer matrix degrada-tion or formation of deposits (fouling/scaling), the feed may flow into the pores in contact with them (surface wetting) [8, 12]. This may initiate internal scaling, which causes wetting of the pores inside the wall and, as a result, in some places, the wetted pores connect with each other, creating channels for feed leakage (partial wetting) [14]. This phenomenon often occurs as a result of salt crystallization on the membrane sur-face [14, 15]. The crystals formed in the pores allow the feed to flow into the next pores. Importantly, this process has been rarely reported in the literature. For this rea-son, several thousand hours of research were carried out in the work, which allowed to indicate the suitability of the tested PTFE membranes for industrial applications.
L77
Therefore, the surface properties of MD membranes applied for the separation of oily water are different from those developed for water desalination, where it is important to increase the hydrophobicity of the membranes. This can be achieved by various methods such as manipulation of pore structure and hydrophobic decorations [26]. Furthermore, wetting of the hydrophobic membranes by the hydrophobic oil should not facilitate the entry of water into the pores. The paper presents a model explaining this phenomenon.
- In the last paragraph of the introduction, please mention the aim of the study.
The last paragraph of the introduction was improved:
L99
Although in recent years the significant achievements in preparing the modified capillary PTFE membranes have been documented [6–8,10,11,33], up to now, for the implementation of the MD process, capillary modules with commercially available membranes are tested. There are only a few manufacturers producing membranes that can be used in the construction of commercial modules. It is essential to mention that in the view of industrial purposes, investigation of the membranes properties during the long-term process are particularly important. However, to the best of the authors knowledge, this issue has not been yet fully evaluated. For this reason, the challenge is to indicate PTFE capillary membranes suitable for the construction of a pilot plant. Worthy of note, testing this type of membranes is essential for the further development of the MD process. Taking the abovementioned into account, herein, the suitability of the capillary PTFE membranes manufactured by an Indian company for the long-term MD processes was assessed. In the present work, the possibility of industrial application of these membranes was assessed by testing their resistance to wetting in over 3500 hours of research with the use of brines contaminated with oil as a feed.
- Please give the temperature as ºC, not K.
It was changed.
- Please present the specifications of M1, M2 and M3 membranes in Materials and Methods.
The article does not mention M1, M2 or M3 membranes - such designations would be a mistake.
One type of capillary PTFE membrane was used in the work. Its properties are presented in Table 1 and in Fig. 3
Membranes were taken from the bundle of membranes and three modules M#1, M#2 and M#3 were made. A module with new membranes was used for each test series.
It is well known that commercial membranes are not always reproducible, so each new module is labelled differently. However, it is also known that in an industrial module, where there are a lot of capillaries, their properties are averaged and there are no significant differences between the modules. This is also shown in Figure 6, where after 150h the MD permeate flux for the tested modules became similar.
Added explanations in the work:
L128
In the current study, the long-term MD experiments with the use of the commercial capillary PTFE membranes manufactured by TECH INC (Tamil Nadu, India) are presented. The membranes were supplied as a bundle of 50 capillaries 30 cm long. The capillaries used for the MD test were pulled randomly from the bundle. The capillary modules were made by gluing the ends of two capillaries to glass tubes (Figure 1b). For the MD study, three similar modules (M#1-3) with an active length of 22 cm (external area 33 cm2) were prepared.
L318
A slightly greater performance was obtained for the M#3 module. Therefore, it can be expected that industrially produced membranes may vary in terms of the permeability achieved. However, several hundred capillaries are mounted in the MD module, which gives an average value and the performance of industrial modules is similar.
- What was the Delta T in the experiments?
The distillate temperature was 22 C and feed temperature was 50-80 ˚C , thus Delta T in the experiments was 28-58 ˚C.
Stream temperatures are given:
L135
During the MD tests, distilled water at the temperature of 22 ±1 ˚C was flowing (150 mL/min) inside the capillaries. The membranes were immersed in a feed heated with RCT Basic magnetic stirrer (IKA-Werke GmbH, Staufen, Germany) to the temperature of 50-80 ±0.5 ˚C.
- What was the effect of the capillary thickness (wall dimension) on heat and mass transfers?
To determine this, the MD model calculations were performed and a new drawing with a description was added to the work. The obtained data indicate that the fluctuations of the wall thickness had a smaller impact than the changes of the internal structure of the membrane wall.
L274
Small differences in the permeate flux could also be due to the wall thickness fluctuations (Figure 4a). Its thickness affects not only mass transport resistance but al-so the amount of QC heat conducted from the feed to the distillate, which is described by the equation (2):
(2)
where is membrane porosity, Sm is membrane thickness, T1 is temperature of evapo-ration surface, T2 is temperature of condensation surface, λV and λPTFE are conductive heat transfer coefficient of vapor (gas phase) and material of membrane matrix (PTFE), respectively. The interfacial temperature T1 and T2 are different from the measured bulk temperatures on the distillate and feed side, and this phenomenon is called the temperature polarization [2,36].
An advantage of submerged MD modules is maintaining a constant feed temper-ature outside the membranes. However, the temperature of distillate stream flowing inside the capillaries decreases along the module length, which decreases the permeate flux. This is mainly due to the increase in the temperature of the condensation surface (T2). Using the MD model presented in [36], the changes in the T2 value for different wall thicknesses were calculated (Figure 9). In each of the analysed cases, the reduction of the wall thickness resulted in a faster increase in the temperature of the boundary layer on the distillate side. However, the permeate flux changes calculated for different feed temperature values did not exceed 5%, which is much less than the differences presented in Figure 7. This shows that the main reason for the differences in the per-meate flux were changes in the wall structure (Figure 4) and to a lesser extent fluctua-tions in its thickness.
Figure 9. Changes in the interfacial surface temperature (T2) on the distillate side calculated along the module length for feed temperature 60˚C and 80˚C. Distillate inlet temperature TD = 22˚C. Membrane wall thickens 0.2 and 0.4 mm.
- Permeate flux values look too low. What was the reason for the low permeate flux?
This performance was due to the rather dense/low porosity wall structure (Figure 4b).
The efficiency of the tested membranes is actually not high. However, as I reviewed the literature, I found that the results obtained for commercial PTFE membranes presented by other authors are similarly small. It is indeed puzzling why commercial membranes have such a low permeate flux, since 2-3 times higher values are presented for PTFE capillaries obtained in laboratory conditions.
- Please explain whether wetting can be reduced with membrane cleaning (Lines 318-319).
This is due to the fact that during the drying of the membrane, the pore wetting brines transforms into salt crystals, which, when the MD resumes, facilitates the flow of the feed into the pores - this re-wets the membrane and reduces permeate flux.
The wetting model (Figure 14) and a description explaining this have been added in the study overview.
L390
Presented and discussed in the literature different intensity of wetting of hydro-phobic membranes by oils, most probably resulted from differences in oil dispersion in the feed. As it was shown in [21,32], the size of oil droplets as well as the stability of the emulsion have a decisive influence on the membranes wetting. This phenomenon is schematically shown in Figure 14a. In the emulsion, large droplets are less stable, thus, they can deposit on the membranes and wet the pores (Figure 14a - pore D), which blocks the vapor flow [25]. Moreover, the decrease in the contact angle value (Figure 10) indicates that the water adsorption on the PTFE capillaries surface occurs during the MD process. A similar phenomenon was found for polypropylene (PP) membranes, which significantly reduced oil fouling [21,32]. As a result, the oil droplets do not block the pores as shown in Figure 14a (pores A).Figure 14. Model of hydrophobic membrane wetting during MD process of brine contaminated with oil. a) influence of pore size and oil droplets diameter, b) effects of internal scaling.
The adsorption of water on the surface of hydrophobic membranes surface significantly depends on the lifetime of the module [43]. Changes in the properties of PTFE membranes, presented in Figure 8, indicate that the membranes stabilized for over 150 h of the MD process. PP membranes showed a similarly long stabilization period, after which a slight increase in the hydrophilicity of the membrane's surface was found, which allowed to reduce oil fouling [21]. Therefore, it is advantageous if the new MD modules are supplied with clean water (without the oil) during the initial 2–3 days, which allows them to obtain a membranes surface with hydrophobic–hydrophilic properties [32]. Omitting the stabilization period meant that the oil droplets completely blocked the pores of PTFE membranes after only 20 h of the MD process [25].
The tested PTFE membranes had large pores on the surface (Figure 5) into which feed can flow (Figure 14a - pore B). As the water evaporates, the oil droplets (pore C) thicken and the salt concentration increases, which reduces the thickness of an electric double layer. The decrease in the double-layer thickness reduces the influence of the electrostatic repulsion and promoting the coalescence [44]. As a result, the oil fills the pore and closes the feed inside it (Figure 14a - pore E). A similar result can also be obtained by the adsorption of a large oil drop on the membrane surface.
Closing even a small amount of brine by oil in the pore can initiate internal scaling and, as a result, cause membrane partial wetting, which is schematically shown in Fig-ure 14b. Due to the water evaporation, the brines become supersaturated which causes the salt crystallization. The fact that the feed has a lower concentration results in os-motic transport through the oil from the feed to the brine in the pores. In the work [45], using a PTFE capillary partially filled with oil, it was shown that oil acts as a semipermeable membrane for water transport when separating brines of different sa-line. Feed transport as a water-in-oil emulsion is also possible. The crystallization of salt in the pore allows the brine to move inside the pores until it fills them completely. Wetting the pores releases the brines into the distillate, resulting in an increase in dis-tillate conductivity. Such a wetting process proceeds very slowly, therefore, during the tests, a greater increase in distillate conductivity occurred only after 400 h MD (Figure 13).
L448
Nonetheless, during the next 72 h of the MD process run, the flux was reduced to 1.4 L/m2h. During the drying of the wetted pores, salt crystals are formed, and consequently, they dissolve when the MD process is resumed. This phenomenon facilitates wetting of the pores and leads to the decrease in the permeate flux. This confirms that drying of the wetted membranes is not an effective method of their regeneration. Finally, a stable increase in the permeate flux to the level of 2.4 L/m2h was obtained after cleaning the membranes with the use of solvent S316, which was applied by Horiba company for very effective oil dissolving. Its hydrophobic properties also cause thorough displacement of brines from the pores. Hence, it can be concluded that removing oil and brines from the membranes wall prevents the pores from re-wetting. Organic solvents can cause membrane swelling, therefore it is necessary to choose solvents suitable for PTFE..
- Wetting includes different forms, surface, partial and full wetting. Please define the wetting observed in the study according to these wetting types.
Added clarifications on what type of wetting was occurring:
L53
However, if hydrophilic groups are formed on the membranes surface as a result of polymer matrix degradation or formation of deposits (fouling/scaling), the feed may flow into the pores in contact with them (surface wetting) [8, 12]. This may initiate internal scaling, which causes wetting of the pores inside the wall and, as a result, in some places, the wetted pores connect with each other, creating channels for feed leakage (partial wetting) [14].
L201
Pores with the largest diameters are wetted the fastest [13], hence their presence may cause partial wetting of the membranes tested [14].
L307
However, the pores inside the membranes wall were not wetted, and as it was presented in the next Section, even for feed contained 300 g NaCl/L the almost 100% retention of salt was obtained, which is characteristic whenever the surface wetting occurs [14].
L341
The decrease in the contact angle value from 134 to 81° shown in Figure 10 indicates that during the long-term period, the surface wetting of the studied membranes occurred. The wall thickness of the membranes was about 400 mm, therefore, despite wetting of the PTFE capillaries surface pores, a layer of pores filled only with the gas phase was still preserved inside their walls. Such a property makes it possible to meet the basic condition for the correct operation of the MD process.
L370
Undoubtedly, this finding indicated a partial wetting, which resulted in a feed leakage into the distillate. However, for the feed concentration of 300 g/L, the high salt retention (> 99%) has been recorded. Probably, only a few of the largest pores (Figure 3) were wetted.
L414
The tested PTFE membranes had large pores on the surface (Figure 5) into which feed can flow (Figure 14a - pore B).
L448
During the drying of the wetted pores, salt crystals are formed, and consequently, they dissolve when the MD process is resumed. This phenomenon facilitates wetting of the pores and leads to the decrease in the permeate flux.
swelling, therefore it is necessary to choose solvents suitable for PTFE.
Reviewer 2 Report
This work, Wettability Studies of Capillary PTFE Membranes Applied for Membrane Distillation, established an interesting work that deals with long-term membrane wetting performance. To better express the novelty and highlight of this work, some issues should be well handled before being further processed. Here are some comments that aim to help improve this work:
(1) First of all, when doing long-term membrane distillation, how to deal with the membrane swelling problem? Since it is announced that 3500 hour MD is conducted.
(2) In the abstract section, the author described some property changes during the MD process, such as permeate flux, wettability transition, distillate conductivity, etc. Nevertheless, the potential working mechanism between structure evolution and property change should be further explored and clarified, which enhances the novelty of this work.
(3) The relationship between wetting performance and MD conditions was discussed in the introduction section. While it should be noted that the wettability evolution is intensively dependent on the pore structure and surface physicochemical property of the membrane, which should be well illustrated in this section with supportive reference (Journal of Power Sources 525(3):231121).
(4) During the 3500h test, how to maintain the test system stability?
(5) The test process of the device displayed in figure1 should be well introduced with a theoretic basis. And how to fix the membrane in this device should be clearly illustrated in this figure.
(6) What is the difference between Figures 2 and 4? Since they are all named SEM images of the capillary PTFE membrane used in the present study.
(7) Discussion on the data should be enhanced to offer more information about the potential affecting mechanism.
Author Response
Dear Reviewer,
I would like to express my sincere gratitude to the Reviewer for the interest in my work and the valuable comments and constructive suggestions.
As indicated below, I have taken into considerations all comments provided by the Reviewer and I have made changes and corrections accordingly to the indications.
Thank you for your time and effort.
Yours sincerely,
Marek Gryta
This work, Wettability Studies of Capillary PTFE Membranes Applied for Membrane Distillation, established an interesting work that deals with long-term membrane wetting performance. To better express the novelty and highlight of this work, some issues should be well handled before being further processed. Here are some comments that aim to help improve this work:
(1) First of all, when doing long-term membrane distillation, how to deal with the membrane swelling problem? Since it is announced that 3500 hour MD is conducted.
It is well known that the water molecule is easily incorporated into the hydrophilic polymer membranes due to the strong affinity between the water molecule and the hydrophilic polymers. Therefore, membrane swelling may occur if such membranes are covered with a hydrophobic layer.
However, there is no water penetration into the membrane matrix if it is made entirely of a highly hydrophobic polymer - such as PTFE. This would have caused the membrane to swell, but in the case studied it was the other way around - the membrane shrunk during the MD (Table 1).
In over 4 years of testing with polypropylene membranes, which are slightly less resistant than PTFE, I have shown that there is no diffusion of water into the membrane, and measurement errors can be the source of reports of diffusion in such polymers. This article has been added to the article:
- Gryta, M. Study of NaCl permeability through a non-porous polypropylene film. Journal of Membrane Science 2016, 504, 66–74, doi./10.1016/j.memsci.2015.12.055
On the other hand, organic solvents can cause swelling of PTFE, hence I added the following information in the work:
- 457
Organic solvents can cause membrane swelling, therefore it is necessary to choose solvents suitable for PTFE.
(2) In the abstract section, the author described some property changes during the MD process, such as permeate flux, wettability transition, distillate conductivity, etc. Nevertheless, the potential working mechanism between structure evolution and property change should be further explored and clarified, which enhances the novelty of this work.
This information has been added:
L17
The mechanism of pore wetting by oil droplets adsorbed on the membrane surface was presented.
L201
Pores with the largest diameters are wetted the fastest [13], hence their presence may cause partial wetting of the membranes tested [14].
L226
It is probable that the tested membranes were annealed too briefly during their production to achieve full thermal stability..
L267
It was shown that not only the pore size, but also the diameter of the inner wall of the fibers has a significant impact on the transport properties of hydrophobic membranes [26]. Therefore, the changes of the virgine membrane (Figure 2b) to a more compact structure (Figure 4b) during the MD had an impact on the permeate flux.
L275
Small differences in permeate flux could also be due to wall thickness fluctuations (Figure 4a). Its thickness affects not only mass transport resistance but also the amount of QC heat conducted from the feed to the distillate, which is described by the equation (2):
|
(2) |
where e is membrane porosity, Sm is membrane thickness, T1 is temperature of evaporation surface, T2 is temperature of condensation surface, λV and λPTFE are conductive heat transfer coefficient of vapor (gas phase) and material of membrane matrix (PTFE), respectively.
L287
Using the MD model presented in [36], the changes in the T2 value for different wall thicknesses were calculated (Figure 9). In each of the analysed cases, the reduction of the wall thickness resulted in a faster increase in the temperature of the boundary lay-er on the distillate side. However, the permeate flux changes calculated for different feed temperature values did not exceed 5%, which is much less than the differences presented in Figure 7. This shows that the main reason for the differences in the permeate flux were changes in the wall structure (Figure 4) and to a lesser extent fluctuations in its thickness.
Figure 9. Changes in the interfacial surface temperature (T2) on the distillate side calculated along the module length for feed temperature 60˚C and 80˚C. Distillate inlet temperature TD = 22˚C. Membrane wall thickens 0.2 and 0.4 mm.
(3) The relationship between wetting performance and MD conditions was discussed in the introduction section. While it should be noted that the wettability evolution is intensively dependent on the pore structure and surface physicochemical property of the membrane, which should be well illustrated in this section with supportive reference (Journal of Power Sources 525(3):231121).
This article has been added and included in the discussion:
[26]. Wang, X.L.; Qu, Z.G.; Lai, T.; Ren, G.F.; Wang, W.K. Enhancing water transport performance of gas diffusion layers through coupling manipulation of pore structure and hydrophobicity, Journal of Power Sources 2022 525, 231121, doi.org/10.1016/j.jpowsour.2022.231121.
L77
Therefore, the surface properties of MD membranes applied for the separation of oily water are different from those developed for water desalination, where it is important to increase the hydrophobicity of the membranes. This can be achieved by various methods such as manipulation of pore structure and hydrophobic decorations [26]. Furthermore, wetting of the hydrophobic membranes by the hydrophobic oil should not facilitate the entry of water into the pores. The paper presents a model explaining this phenomenon.
L267
It was shown that not only the pore size, but also the diameter of the inner wall of the fibers has a significant impact on the transport properties of hydrophobic membranes [26]. Therefore, the changes of the virgine membrane (Figure 2b) to a more compact structure (Figure 4b) during the MD had an impact on the permeate flux.
(4) During the 3500h test, how to maintain the test system stability?
We rely on the effectiveness of the electronic control systems that have the apparatus used to build the experimental installation. The air temperature in the laboratory changes (slightly), so the feed temperature control system (stirrer with heating and temperature measurement) automatically regulates the heating power and, as shown by daily control measurements, the changes in TF did not exceed 0.5 K from the assumed value. The cooling system, after adjusting the tap water flow valve, maintained the distillate temperature in the range of 293+/- 1 K. The volume of the feed was 4.4 L, and the daily losses did not exceed 0.1 L, which even for a concentration of 300 g/L would give an increase in the concentration of the feed after the MD day to 306 g/L, which by daily addition of distilled water returned to 300 g/L.
The paper presents the results for over 3,500 hours of testing, but similar installations have also proven themselves in tests lasting over 10,000 hours.
In Experimental, descriptions of the accuracy of maintained parameters have been added:
L135
During the MD tests, distilled water at the temperature of 22 ±1 ˚C was flowing (150 mL/min) inside the capillaries. The membranes were immersed in a feed heated with RCT Basic magnetic stirrer (IKA-Werke GmbH, Staufen, Germany) to the temperature of 50-80 ±0.5 ˚C.
L152
The volume losses in the feed tank were every day replenished with distilled water.
Figure 1 -T – thermometer connected with heat power regulator
(5) The test process of the device displayed in figure1 should be well introduced with a theoretic basis. And how to fix the membrane in this device should be clearly illustrated in this figure.
Figure 1 shows a typical MD setup for the study of submerged MD modules well known and used in publications presented by various MD authors.
A picture of the installation has been added (Figure 1a) showing how the membranes were installed. A description of the module implementation and the basics of measurements have also been added.
Figure 1. Experimental MD set-up. a) Installation scheme: 1 – submerged module, 2 – feed tank, 3 – magnetic stirrer with heating plate, 4 – balance, 5 – peristaltic pump, 6 – distillate tank, T – thermometer connected with heat power regulator. b) submerged module connected with distillate line.
L130
The membranes were supplied as a bundle of 50 capillaries 30 cm long. The capillaries used for the MD test were pulled randomly from the bundle. The capillary modules were made by gluing the ends of two capillaries to glass tubes (Figure 1b). For the MD study, three similar modules (M#1-3) with an active length of 22 cm (external area 33 cm2) were prepared.
L142
As a result of pore wetting, there is the possibility of wetting part of the wall through which salt diffuses from the feed into the distillate. In order to detect it, the changes in the distilled conductivity were investigated examined. In addition, partial wetting and oil adsorption reduce feed evaporation surfaces, which reduces the permeate flux. To assess the intensity of these phenomena, the MD tests were run continuously. Once a day, the distilled conductivity and the permeate flux were determined. The permeate flux (L/m2h) was calculated as the mean of the distillate volume collected over 24 h, according to the following equation:
J=dV/A t (1)
where dV (L) is an increase in the volume of distillate at time t (24h), and A (m2) is the module area. The volume losses in the feed tank were every day replenished with distilled water.
(6) What is the difference between Figures 2 and 4? Since they are all named SEM images of the capillary PTFE membrane used in the present study.
I added a description that distinguishes these samples:
Figure 2. SEM images of the virgin capillary PTFE membrane: (a) Capillary cross-section; (b) Wall cross-section; (c) External surface; (d) Internal surface.
Figure 4. SEM images of the capillary PTFE membrane collected from M#1 module after 2000 h of the MD process run (TF=60 ˚C): (a) Capillary cross-section; (b) Wall cross-section; (c) External surface; (d) Internal surface.
(7) Discussion on the data should be enhanced to offer more information about the potential affecting mechanism.
New information has been added:
L201
Pores with the largest diameters are wetted the fastest [13], hence their presence may cause partial wetting of the membranes tested [14].
L226
It is probable that the tested membranes were annealed too briefly during their production to achieve full thermal stability.
L267
It was shown that not only the pore size, but also the diameter of the inner wall of the fibers has a significant impact on the transport properties of hydrophobic membranes [26]. Therefore, the changes of the virgine membrane (Figure 2b) to a more compact structure (Figure 4b) during the MD had an impact on the permeate flux.
L274
Small differences in the permeate flux could also be due to the wall thickness fluctuations (Figure 4a). Its thickness affects not only mass transport resistance but also the amount of QC heat conducted from the feed to the distillate, which is described by the equation (2):
|
(2) |
where e is membrane porosity, Sm is membrane thickness, T1 is temperature of evaporation surface, T2 is temperature of condensation surface, λV and λPTFE are conductive heat transfer coefficient of vapor (gas phase) and material of membrane matrix (PTFE), respectively. The interfacial temperature T1 and T2 are different from the measured bulk temperatures on the distillate and feed side, and this phenomenon is called the temperature polarization [2,36].
An advantage of submerged MD modules is maintaining a constant feed temperature outside the membranes. However, the temperature of distillate stream flowing inside the capillaries decreases along the module length, which decreases the permeate flux. This is mainly due to the increase in the temperature of the condensation surface (T2). Using the MD model presented in [36], the changes in the T2 value for different wall thicknesses were calculated (Figure 9). In each of the analysed cases, the reduction of the wall thickness resulted in a faster increase in the temperature of the boundary layer on the distillate side. However, the permeate flux changes calculated for different feed temperature values did not exceed 5%, which is much less than the differences presented in Figure 7. This shows that the main reason for the differences in the permeate flux were changes in the wall structure (Figure 4) and to a lesser extent fluctuations in its thickness.
Figure 9. Changes in the interfacial surface temperature (T2) on the distillate side calculated along the module length for feed temperature 60˚C and 80˚C. Distillate inlet temperature TD = 22˚C. Membrane wall thickens 0.2 and 0.4 mm.
L318
A slightly greater performance was obtained for the M#3 module. Therefore, it can be expected that industrially produced membranes may vary in terms of the permeability achieved. However, several hundred capillaries are mounted in the MD module, which gives an average value and the performance of industrial modules is similar.
L341
The decrease in the contact angle value from 134 to 81° shown in Figure 10 indicates that during the long-term period, the surface wetting of the studied membranes occurred. The wall thickness of the membranes was about 400 mm, therefore, despite wetting of the PTFE capillaries surface pores, a layer of pores filled only with the gas phase was still preserved inside their walls. Such a property makes it possible to meet the basic condition for the correct operation of the MD process.
A concentration of 300 g NaCl/L is approaching saturation, which may cause scaling. In this case, the separation possibility of the saturated solutions can be achieved by reducing the salt concentration in the feed using a crystallizer attached to the MD installation (membrane distillation crystallizer) [15,37].
L370
Undoubtedly, this finding indicated a partial wetting, which resulted in a feed leakage into the distillate. However, for the feed concentration of 300 g/L, the high salt retention (> 99%) has been recorded. Probably only a few of the largest pores (Figure 3) were wetted.
L390
Presented and discussed in the literature different intensity of wetting of hydrophobic membranes by oils, most probably resulted from differences in oil dispersion in the feed. As it was shown in [21,32], the size of oil droplets as well as the stability of the emulsion have a decisive influence on the membranes wetting. This phenomenon is schematically shown in Figure 14a. In the emulsion, large droplets are less stable, thus, they can deposit on the membranes and wet the pores (Figure 14a - pore D), which blocks the vapor flow [25]. Moreover, the decrease in the contact angle value (Figure 10) indicates that the water adsorption on the PTFE capillaries surface occurs during the MD process. A similar phenomenon was found for polypropylene (PP) membranes, which significantly reduced oil fouling [21,32]. As a result, the oil droplets do not block the pores as shown in Figure 14a (pores A).
Figure 14. Model of hydrophobic membrane wetting during MD process of brine contaminated with oil. a) influence of pore size and oil droplets diameter, b) effects of internal scaling.
The adsorption of water on the hydrophobic membranes surface significantly depends on the module lifetime [43]. Presented in Figure 8 changes in the properties of PTFE membranes, indicate that the membranes stabilized for over 150 h of the MD process run. The PP membranes showed a similarly long stabilization period, after which a slight increase in the hydrophilicity of the membrane's surface was found, which allowed to reduce the oil fouling [21]. Therefore, it is advantageous if the new MD modules are supplied with clean water (without the oil) during the initial 2–3 days, which allows them to obtain a membranes surface with hydrophobic–hydrophilic properties [32]. Omitting the stabilization period meant that the oil droplets completely blocked the pores of PTFE membranes after only 20 h of the MD process [25].
The tested PTFE membranes had large pores on the surface (Figure 5) into which feed can flow (Figure 14a - pore B). As the water evaporates, the oil droplets (pore C) thicken and the salt concentration increases, which reduces the thickness of an electric double layer. The decrease in the double-layer thickness reduces the influence of the electrostatic repulsion and promoting the coalescence [44]. As a result, the oil fills the pore and closes the feed inside it (Figure 14a - pore E). A similar result can also be obtained by the adsorption of a large oil drop on the membrane surface.
Closing even a small amount of brine by oil in the pore can initiate internal scaling and, as a result, cause membrane partial wetting, which is schematically shown in Figure 14b. Due to the water evaporation, the brines become supersaturated which causes the salt crystallization. The fact that the feed has a lower concentration results in osmotic transport through the oil from the feed to the brine in the pores. In the work [45], using a PTFE capillary partially filled with oil, it was shown that oil acts as a semipermeable membrane for water transport when separating brines of different saline. Feed transport as a water-in-oil emulsion is also possible. The crystallization of salt in the pore allows the brine to move inside the pores until it fills them completely. Wetting the pores releases the brines into the distillate, resulting in an increase in distillate conductivity. Such a wetting process proceeds very slowly, therefore, during the tests, a greater increase in distillate conductivity occurred only after 400 h MD (Figure 13).
L448
Nonetheless, during the next 72 h of the MD process run, the flux was reduced to 1.4 L/m2h. During the drying of the wetted pores, salt crystals are formed, and consequently, they dissolve when the MD process is resumed. This phenomenon facilitates wetting of the pores and leads to the decrease in the permeate flux. This confirms that drying of the wetted membranes is not an effective method of their regeneration. Finally, a stable increase in the permeate flux to the level of 2.4 L/m2h was obtained after cleaning the membranes with the use of solvent S316, which was applied by Horiba company for very effective oil dissolving. Its hydrophobic properties also cause thorough displacement of brines from the pores. Hence, it can be concluded that removing oil and brines from the membranes wall prevents the pores from re-wetting. Organic solvents can cause membrane swelling, therefore it is necessary to choose solvents suitable for PTFE.
Round 2
Reviewer 1 Report
I have carefully reviewed the R1 version of the manuscript. The authors have responded to my comments by addressing my major concerns and have improved the manuscript accordingly. I have no comment at this stage.
Reviewer 2 Report
The revised manuscript in current status has been greatly improved with enhanced presentation in the novelty and highlights, and could be further processed.